analytical chemistry/organic chemistry/ chemical physics

pyrene, CMC determination, measurement condition, sample preparation method, surfactant

**Author for correspondence:**
Qiuhua Zhu
e-mail: zhuqh@smu.edu.cn

This article has been edited by the Royal Society of Chemistry, including the commissioning, peer review process and editorial aspects up to the point of acceptance.

[†]These authors contributed to this manuscript equally.

# Influence factors on the critical micelle concentration determination using pyrene as a probe and a simple method of preparing samples

Hao Li[†], Danna Hu[†], Feiqing Liang, Xiaowei Huang and Qiuhua Zhu

Guangdong Provincial Key Laboratory of New Drug Screening, School of Pharmaceutical Sciences, Southern Medical University, 1838 Guangzhou Avenue North, Guangzhou 510515, People's Republic of China

QZ, 0000-0003-3731-4487

The critical micelle concentration (CMC) is an important parameter of widely used surfactants and needs to be measured in the application and development of surfactants. Fluorometric method is a widely used method determining CMC values owing to the advantages of highly sensitivity, fast response and wide application range. There are two common methods (I and II) of preparing samples for CMC fluorometric determination. In the process of developing CMC probes with aggregation-induced emission (AIE) characteristics, we found that methods I and II were not suitable for CMC probes with AIE charateristics and developed a new sample preparation method (III), which is not only suitable for CMC probes with AIE characteristic but also decreases operation procedures and errors owing to omitting the addition of micro amount of dyes into each sample. To ascertain if method III is also suitable for other CMC probes without AIE characteristics, the CMC values of surfactants were determined by fluorometric method using widely used pyrene without AIE charateristic as probe and methods I–III to prepare samples. The obtained experimental results proved that method III not only was suitable for preparation of samples for CMC determination of surfactants using pyrene as probe but also led to the least average deviation (methods I–III led to ±0.13, ±0.34 and ±0.05 mM deviation for the CMC determination of sodium dodecyl sulfate (SDS), respectively). The CMC determination using pyrene as probe is based on its change in the ratio $(I_{FIII}/I_{FI})$ of its emission peaks I and III with surfactant concentration. Unexpectedly, it was found that the $I_{FIII}/I_{FI}$

value of pyrene in surfactant solutions is sensitive to the measurement conditions changing exciting light energy, such as slit widths and sample-measured number. In addition, it was found that surfactant SDS or cetrimonium bromide from different suppliers not only has significantly different CMC values but also leads to very different $I_{FIII}/I_{FI}$ values of pyrene in a certain concentration of surfactant, which can be used as a simple method to distinguish the same surfactant with different CMC values.

# 1. Introduction

Critical micelle concentration (CMC) is an important parameter of surfactants with wide applications [1–5]. The CMC values of surfactants not only relate to their molecular structures but also are sensitive to environments and relate to suppliers [6,7]. Therefore, their CMC values need to be measured in their practical applications and research [8–13]. Among current methods for CMC determination, the fluorescence method based on the fluorescent change of organic probes attracts great attention owing to its high sensitivity and fast response [6,8,14–17]. However, the fluorescent changes are usually invisible or not sharp about CMC and hence a series of samples containing different concentrations of surfactant and a certain amount of probe need to be prepared and measured by a fluorospectrophotometer. There are two general methods (methods I and II) for preparing samples. If the fluorescence changes are very sharp and visible about/at CMC, the CMC values will be directly observed. In 2011 [18], we developed an efficient five-component reaction for the synthesis of a novel series of C6-unsubstituted tetrahydropyrimidines (THPs) with strong aggregation-induced emission (AIE) characteristics, that is, no emission in solution but strong fluorescence in aggregates. AIE characteristic, found and termed by Tang's group [19], resolves the thorny aggregation-caused quenching problem of conventional fluorophores and has shown great advantages in wide areas [20], such as AIEgens-containing copolymers and their applications [21–25]. The characteristics of THPs—completely no emission in surfactant micelles but strong AIE in dilute surfactant solutions—let us develop them as unique sensitive and visible fluorescence-turn-on (showing the strongest fluorescence at CMC) probes for CMC (the reported CMC probes based on fluorescence intensity change show fluorescence-turn-off change at CMC, that is, show the weakest fluorescence at CMC) [26,27]. Recently, we found that one of the THPs could be used as an excellent indicator for CMC titration and realized simple, sample- and time-saving CMC titration for different kinds of surfactants for the first time [28].

In the process of developing highly sensitive fluorescence-turn-on probes (THPs) for CMC determination, we found that methods I and II were not suitable for THPs with the characteristics of aggregation-induced emission (AIE) in dilute surfactant solutions and no emission in surfactant micelles and developed a new method (method III) [26]. Method III is not only suitable for THPs but also has the advantages of less operation procedures and errors owing to omitting the addition of micro amount of dye into each sample. We wondered whether method III was also suitable for other CMC probes without AIE characteristics. Considering that pyrene is the most used fluorescent probe for CMC determination [15,29–32], we prepared samples by methods I–III and studied the factors influencing CMC determination using pyrene as probe in detail. The CMC determination using pyrene as probe is based on the linear relationship between the surfactant concentration and the ratio ($I_{FIII}/I_{FI}$) of its fluorescence intensities at the peaks I and III. Unexpectedly, we found that the $I_{FIII}/I_{FI}$ value of pyrene in surfactant solutions is unusually sensitive to the measurement conditions changing exciting light energy. In addition, we found that some of the surfactants from different suppliers not only had different CMC values but also led to significantly different $I_{FIII}/I_{FI}$ values in surfactant solutions, which can be used as a very simple and useful method to distinguish the same surfactant with different CMC values.

# 2. Experimental section

## 2.1. Materials and instruments

All chemicals used in this paper were obtained from commercial suppliers and used without further purification. Surfactant CHAPS was purchased from Energy Chemical; sodium dodecyl sulfate (SDS) was purchased from Guangzhou Weijia Technology Co., Ltd, SERVA Electrophoresis GmbH and Shanghai Meryer Chemical Technology Co., Ltd; cetrimonium bromide (CTAB) was purchased from

Tianjin Damao Chemical Reagent Factory and Aladdin; Triton X-100 was purchased from Aladdin; and BS-12 was purchased from Shanghai Shengxuan Biology Chemical Co., Ltd (for the molecular structures of these surfactants, see the electronic supplementary material). All measurements were carried out at $25 \pm 1°C$. Water was purified via deionization and filtrated by the Millipore purification to resistivity higher than $18\,M\Omega\,cm^{-1}$. Excitation and emission spectra were determined by FluoroMax-4 spectrofluorophotometer (unnoted, emitted at 373 nm and excited at 334 nm, excitation and emission slit widths: 2 and 2 nm or 3 and 3 nm, and samples were determined immediately after preparation).

## 2.2. Preparation of pyrene ethanol stock solution (0.5 mM)

About 10.1 mg of pyrene ($M_r = 202.3$) and about 80 ml of ethanol were added into a 100 ml volumetric flask, shaking well for dissolution, then filling the flask to the mark with ethanol.

## 2.3. Preparation of samples by method I

A certain amount of surfactant stock solution and pyrene stock solution were added into a 100 ml volumetric flask, shaking well and keeping at least for 30 min before filling the flask to the mark with water to prepare a concentrated surfactant solution (about 2CMC) with a certain concentration of pyrene (0.2–1 µM). Then different volumes of the concentrated surfactant solution were added into different 5 ml volumetric flasks, filling these flasks to the mark with water containing the same concentration of pyrene as that in the concentrated surfactant solution.

## 2.4. Preparation of samples by method II

Different volumes of surfactant stock solutions and 2 ml of pyrene-saturated water solution were added into different 5 ml volumetric flasks, shaking well and keeping at least for 30 min before filling these flasks to the mark with water.

## 2.5. Preparation of samples by method III

Method III was reported in our previous work [26]. Generally, a concentrated surfactant solution (about 2CMC) with a certain concentration of pyrene (0.5–1.5 µM) was prepared firstly as method I. Then different volumes of the concentrated surfactant solution containing certain amount of pyrene were added into different 5 ml volumetric flasks, filling these flasks to the mark with water.

# 3. Results and discussion

## 3.1. Influence factors on critical micelle concentration determination of sodium dodecyl sulfate using pyrene as probe and method I for sample preparation

To study the factors influencing CMC determination using pyrene as probe, the CMC value of commonly used anionic surfactant SDS was determined under different conditions, and samples were prepared by commonly used method I, that is, a concentrated SDS solution (10 mM) containing a certain amount of pyrene (0.5 µM) was diluted to different concentrations of SDS solutions with water containing 0.5 µM of pyrene. To study the influence of sample-kept time, the prepared samples were measured instantly, 0.5 and 24 h, respectively, by a fluorospectrometer. The experimental results indicate that sample-kept time shows no influence on the excitation and emission spectra of pyrene (figure 1a–c), and the determined CMC average value and standard deviation of SDS is $6.53 \pm 0.12$ mM. The standard deviation is much smaller than the reported $\pm 0.4$ mM using pyrene as probe [6]. There are two intersections in figure 1d. The first intersection rather than the second intersection is corresponding to CMC value. This is because, with increase of SDS concentration, below CMC, SDS exists in monomers; at CMC, the concentration of SDS in monomers reaches the maximum and the micelles begin to form; at the same time, with increase of SDS concentration, below CMC, the $I_{FIII}/I_{FI}$ value increases smoothly owing to the influence of SDS monomers; from CMC to the second intersection, the $I_{FIII}/I_{FI}$ value increases sharply owing to the fast transfer of pyrene from solution into micelles; from the second intersection, the $I_{FIII}/I_{FI}$ value increases smoothly again because the pyrene concentrations in SDS micelle and solution phases, respectively, are in an equilibrium and hence pyrene slowly transfers from solution

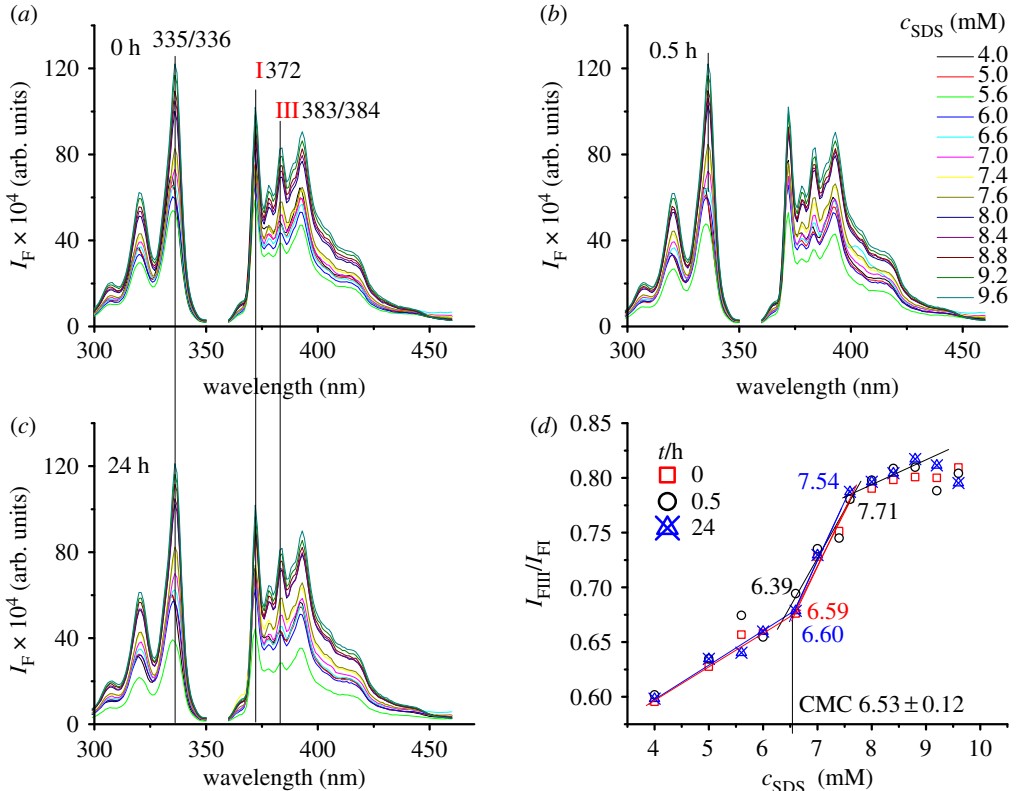

**Figure 1.** Influence of sample-kept time on the CMC determination of SDS. (*a–c*) The excitation (left) and emission (right) spectra of pyrene (0.5 µM) in SDS solutions with different concentrations (4–10 mM) kept for 0, 0.5 and 24 h, respectively; (*d*) relationship between SDS concentration and the $I_{FIII}/I_{FI}$ value of pyrene in (*a–c*).

into micelles; from SDS concentration higher than 9 mM, the $I_{FIII}/I_{FI}$ value keeps almost the same because the concentration of pyrene in solution is very low and the amount of pyrene transferring from solution into micelles is too small to cause the change in the $I_{FIII}/I_{FI}$ value.

Since sample-kept time shows no influence on CMC determination, the samples prepared by method I were measured immediately to study other factors influencing the CMC determination of SDS using pyrene as probe. The study on the influence of the concentration ($c_{pyr}$) of pyrene is shown in figure 2*a* and electronic supplementary material, figure S1. The experimental results indicate that when $c_{pyr}$ is 0.2, 0.5 and 1.0 µM (limited by solubility in water, higher $c_{pyr}$ was not studied), the average value and standard deviation of these determined CMC values is 6.66 ± 0.18 mM, with the deviation smaller than the reported one (±0.4 mM) [6]. This proves that in the range of 0.2–1.0 µM, pyrene shows no influence on CMC value. This was further demonstrated by almost the same CMC values (7.15 and 7.21 mM) of SDS in the presence and absence of pyrene (0.5 µM) determined by conductive method (the average deviation is ±0.1 mM for the CMC values of SDS determined by conductive method [6]) (electronic supplementary material, figure S2). It is worth mentioning that although pyrene shows no influence on the CMC determination of SDS, the CMC value (6.64 mM, electronic supplementary material, figure S2b) determined by fluorometric method using pyrene as probe is lower than that (7.15 mM, electronic supplementary material, figure S2c) determined by conductive method. This case is the same as that reported [6].

Among lots of the obtained $I_{FIII}/I_{FI}$ values, some of them are significantly higher than normal values, which puzzled us. After considerable efforts to explore the factors causing the abnormal change in $I_{FIII}/I_{FI}$, we finally found that the $I_{FIII}/I_{FI}$ value of pyrene increased significantly when enlarging the slit widths of the fluorospectrometer but the determined CMC value is within the measurement error range (figure 2*b*). Since the increase of the fluorescence intensity caused by pyrene concentration almost did not cause the change in the $I_{FIII}/I_{FI}$ value (figure 2*a*; electronic supplementary material, figure S1), we deduced that it was the enhancement of the light energy exciting pyrene that caused the increase of the $I_{FIII}/I_{FI}$ value when enlarging the slit widths of the fluorospectrometer, and if the number ($N$) measuring the fluorescence spectra of pyrene increased, which means increasing the energy exciting pyrene, the $I_{FIII}/I_{FI}$ value will also increase. This was proved by the experimental results; the $I_{FIII}/I_{FI}$ value increased

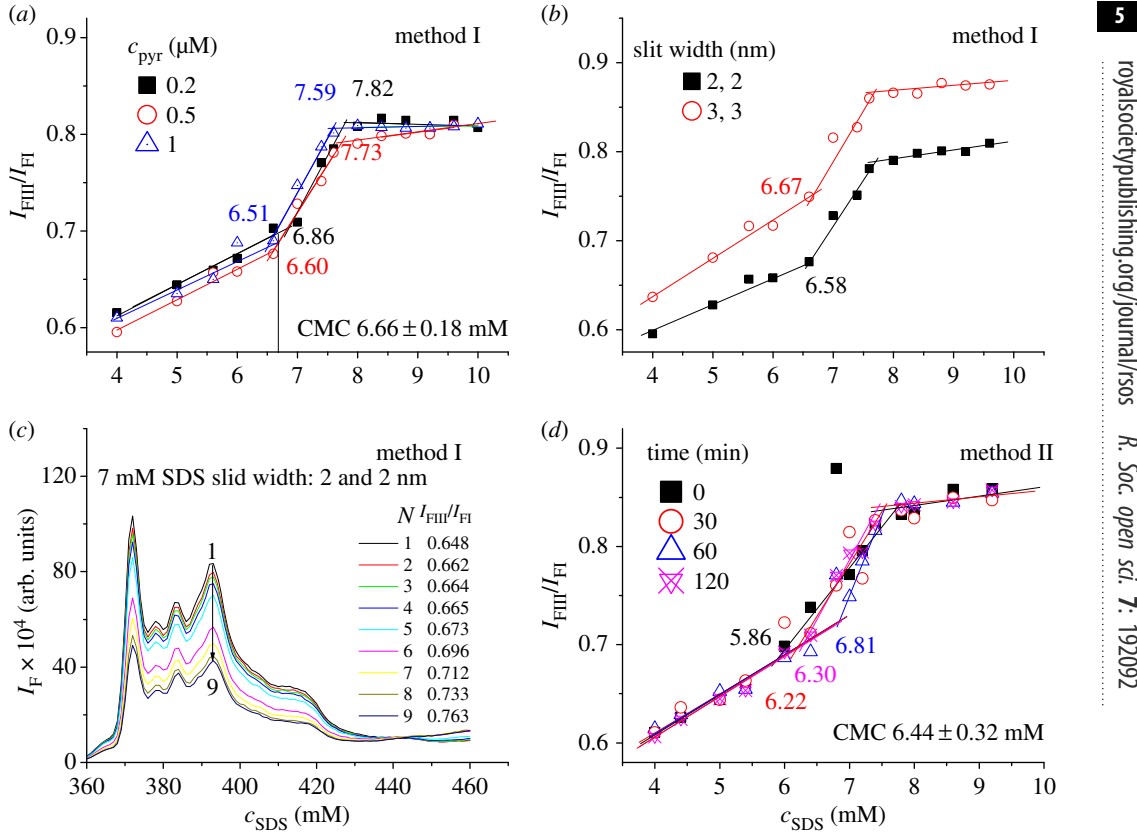

**Figure 2.** (*a*–*d*) Influences of pyrene concentration ($c_{pyr}$), fluorospectrometer slit widths, sample-measured number (*N*), and sample-kept time, respectively, on the CMC determination of SDS using pyrene as probe. Samples were prepared by method I (*a*–*c*) ($c_{pyr}$ is 0.5 µM) or method II (*d*).

from 0.648 to 0.763 when *N* increased from 1 to 9 times (figure 2*c*). These results well explain why some of the obtained $I_{FIII}/I_{FI}$ values are abnormally high.

The above results indicate that in a suitable concentration of pyrene (0.2–1.0 µM), the CMC average value and standard deviation of all determined SDS values in figures 1*d*, 2*a* and 2*b* is 6.60 ± 0.13 mM.

## 3.2. Critical micelle concentration determination of sodium dodecyl sulfate via samples prepared by method II

With the above optimized conditions for CMC determination using pyrene as probe, a series of samples with different concentrations of SDS and 2 ml of pyrene-saturated water solution were prepared by method II and measured by a fluorospectrometer immediately. Unexpectedly, the CMC value determined from the samples prepared by method II (the CMC value determined from samples kept 0 min in figure 2*d*) is much lower (5.86 mM) than that (6.60 ± 0.13 mM) determined from samples prepared by method I. After studying the influence factors on the CMC values, we found that after different volumes of the concentrated surfactant solution and 2 ml of pyrene-saturated water solution were added into different 5 ml volumetric flasks, the mixtures should be shaken well and kept at least 30 min before filling these flasks to the mark with water (figure 2*d*; electronic supplementary material, figure S4). The CMC values determined from the samples kept for 30 and 120 min are almost the same (6.22 and 6.30 mM) although that determined from the samples kept 60 min is significantly higher (6.81 mM). These results indicate that samples could be measured after keeping 30 min and that the $I_{FIII}/I_{FI}$ values from the first interaction to the second interaction are not very stable and easily influenced by determined conditions. The average CMC value and deviation of SDS determined from the samples prepared by method II is 6.44 ± 0.32 mM, with larger standard deviation than that determined from the samples prepared by method I.

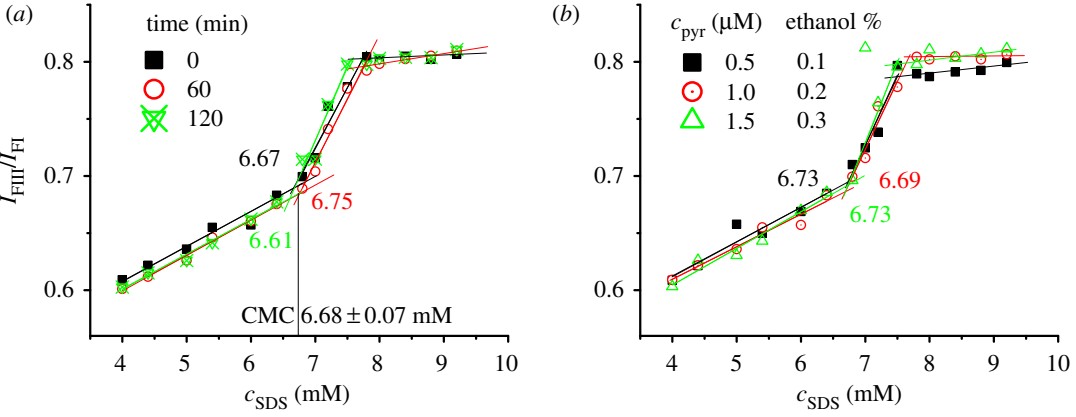

**Figure 3.** Influences of sample-kept time (*a*) and pyrene concentration (*b*) on CMC determination of SDS using pyrene as a probe. Samples were prepared by method III and the concentration of pyrene in (*a*) was 1 µM in 10 mM SDS solution.

**Table 1.** CMC values of different kinds of surfactants determined from the samples prepared by methods I–III.

| surfactant | CMC/mM | | | cond[a] | reported[b] |
| | method I | method II | method III | | |
|---|---|---|---|---|---|
| SDS[c] | $6.60 \pm 0.13$ | $6.44 \pm 0.32$ | $6.70 \pm 0.05$ | 7.21 | 2.9 [7] to $7.9 \pm 0.4$ [6] |
| SDS[d] | 5.36 | 5.35 | 5.39 | 6.22 | |
| CTAB[e] | 0.64 | | 0.62 | | $0.7 \pm 0.2$ [6] to 0.88 [26] |
| CTAB[f] | | | 0.80 | | |
| CHAPS | 7.01 | | 7.09 | | 7.4 [33] to 7.5 [26] |
| BS-12 | 2.20 | | 2.24 | | 1.1 [34][g] |
| Triton X-100 | 0.18 | | 0.16 | | 0.08 [7] to $0.37 \pm 0.09$ [6] |

[a]conductive method.
[b]CMC value determined using pyrene as probe.
[c]Weijia or SERVA reagent.
[d]Meryer reagent.
[e]Damao reagent.
[f]Aladdin reagent.
[g]CMC value determined by surface tension method.

## 3.3. Critical micelle concentration determination of sodium dodecyl sulfate via samples prepared by method III

To evaluate whether method III (diluting concentrated surfactant solution containing a certain amount of probe into a series of samples containing different concentrations of SDS and pyrene with pure solvent) is suitable for preparation of samples using pyrene as CMC probe, the factors influencing CMC determination of SDS were investigated in detail. The obtained experimental results indicate that the samples prepared by method III can be determined immediately (figure 3*a*) and pyrene shows no influence when $c_{pyr}$ is 0.5–1.5 µM in the concentrated SDS solution (figure 3*b*). The excitation and emission spectra of pyrene in these determined samples are shown in electronic supplementary material, figure S5 and S6. The average value and standard deviation of the six determined CMC values in figure 3 is $6.70 \pm 0.05$ mM, with much smaller deviation than those measured from the samples prepared by methods I and II ($\pm 0.13$ and 0.32 mM). According to the experimental results in figure 3*b*, one can deduce that ethanol shows no influence on CMC determination when the concentration of ethanol in concentrated SDS solution is lower than 0.3%. This is because the increase of $c_{pyr}$ from 1 µM to 1.5 µM means the increase of the concentration of ethanol from 0.2% to 0.3%, but the determined CMC values are almost the same (figure 3*b*).

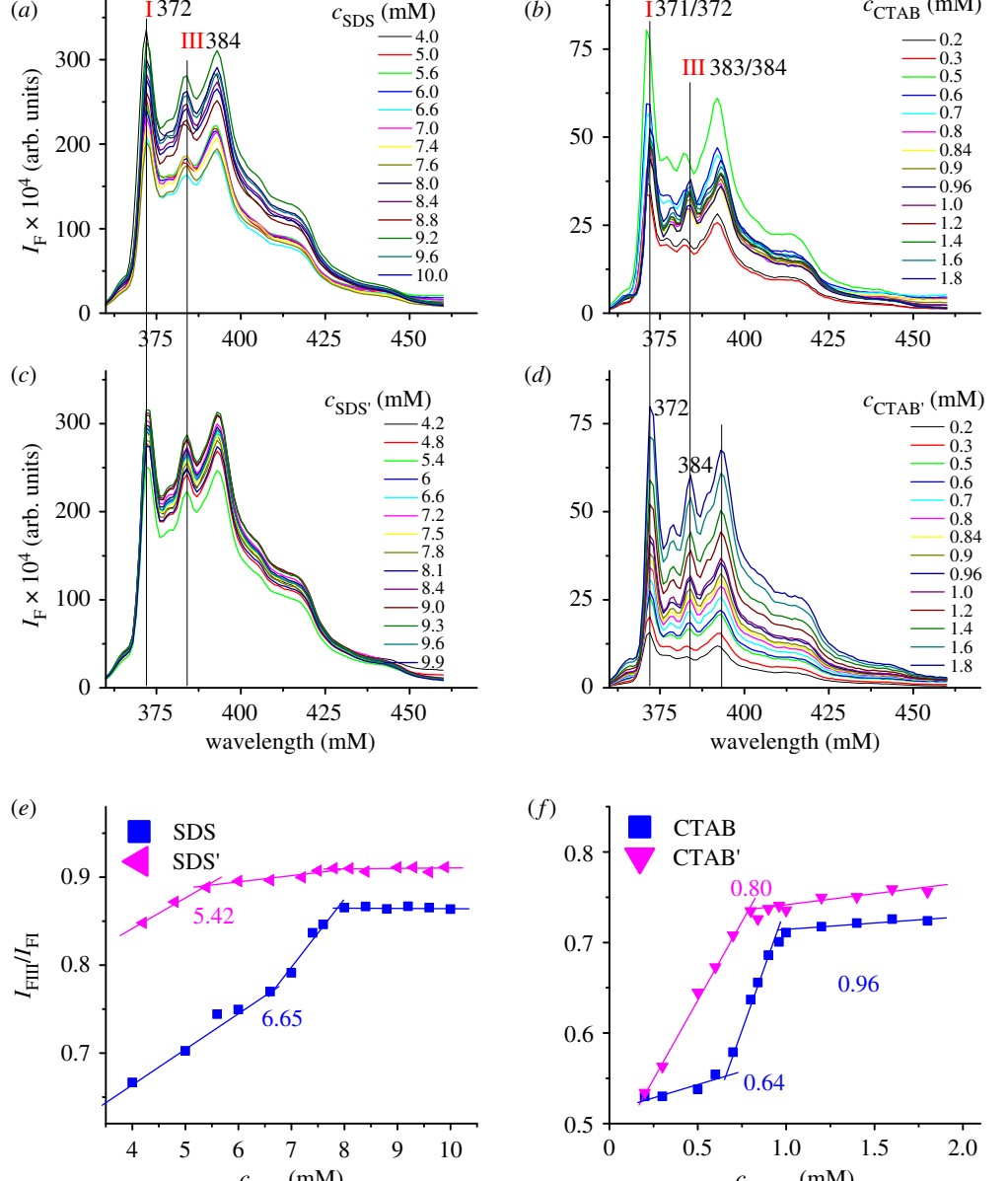

**Figure 4.** Influences of different sources of SDS and CTAB on the fluorescent properties of pyrene and their CMC values. (*a–d*) Emission spectra of pyrene in SDS, SDS', CTAB and CTAB' solutions, respectively. (*e,f*) The relationship between the $I_{FIII}/I_{FI}$ value and the concentration of SDS/SDS' and CTAB/CTAB', respectively. The samples were prepared by method I (*a–c*) or method III (*d*).

## 3.4. Critical micelle concentration determination of other kinds of surfactants from the samples prepared by method I and III

To further identify whether method III was suitable for preparing samples using pyrene as CMC probe, the CMC values of cationic surfactant CTAB, zwitterionic surfactant CHAPS, non-ionic surfactants Triton X-100 and BS-12 were determined from samples prepared via method I and III. The excitation and emission spectra of pyrene in the samples prepared by method I/III are shown in electronic supplementary material, figure S7 and S8/S9 and S10. The determined CMC values of different kinds of surfactants are shown in table 1. The CMC values determined from the samples prepared by method I and III are almost the same except SDS. These results prove that method III is suitable for preparing samples for CMC determination of different kinds of surfactants using pyrene as a probe.

## 3.5. Influence of different sources of sodium dodecyl sulfate and cetrimonium bromide on their critical micelle concentration values and the $I_{FIII}/I_{FI}$ values

Interestingly, we found that SDS and CTAB from different suppliers not only have different CMC values (table 1) but also lead to different $I_{FIII}/I_{FI}$ value (figure 4$e$ and $f$), especially the $I_{FIII}/I_{FI}$ values at 4 mM SDS and 0.5 mM CTAB, which can be used as a simple method to distinguish the surfactant with different CMC values. The different CMC values of SDS or CTAB might be caused by purity [6]. In addition, the outlines or wavelengths of peak I and III are also different (comparing the tops of peak I in figure 4$a$ and $b$, and the wavelengths of peaks I and III in figure 4$c$ and $d$). The CMC determination of SDS' by conductive method and fluorometric method using pyrene as probe (samples were prepared by methods I–III) are shown in electronic supplementary material, figure S11–14. From the emission spectra of pyrene in figure 4, one might notice that with change in surfactant concentration, the change in the fluorescence intensity of pyrene is irregular in the samples prepared by methods I and II but regular in the samples prepared by method III. This is because the addition of micro amount of pyrene into each sample, which is omitted in method III but needed in methods I and II, will inevitably cause different errors in pyrene concentration and hence lead to irregular change in the fluorescence intensity of pyrene.

## 4. Conclusion

We investigated the factors influencing CMC determination using pyrene as a probe and compared methods I–III of preparing samples for CMC determination. Methods I and II, two commonly used methods, are only suitable CMC probes without AIE characteristics, and method III, developed by us for CMC probes with AIE characteristics, has advantages of less operation procedures and errors owing to omitting the addition of micro amount of probe into each sample. It was found as the following: (i) Method III, omitting the addition of micro amount of CMC dye into each sample and hence decreasing operation procedures and errors, proved not only suitable for preparing samples using pyrene without AIE characteristics as CMC probe but also the best (with the simplest procedures and smallest standard deviation). (ii) In the process of preparing samples, the mixture of concentrated surfactant solution and pyrene must be kept for at least 30 min before it is diluted. (iii) The $I_{FIII}/I_{FI}$ value of pyrene is unusually sensitive to the measurement conditions relating to the light energy exciting pyrene such as slit widths and measurement number. (iv) SDS and CTAB from different suppliers not only had different CMC values but also led to the $I_{FIII}/I_{FI}$ value of pyrene in 4 mM SDS and 0.5 mM CTAB solutions being very significant, by which one can simply distinguish SDS or CTAB from different suppliers.

Data accessibility. Data have been uploaded as part of the electronic supplementary material.

Authors' contributions. H.L. gave substantial contributions to acquisition, analysis and interpretation of data; D.H. conducted part of experiments and took part in the analysis and interpretation of data and article writing; X.H. and F.L. took part in acquisition, analysis and interpretation of data; Q.Z. contributed to conception, design, analysis and interpretation of data, and article writing.

Competing interests. We declare we have no competing interests.

Acknowledgements. We are very grateful for financial support from the Special Fund for Scientific and Technological Innovation and Cultivation of Guangdong University Students (pdjh2019b0102).

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
