## [Reviewer comments · Royal Society Open Science]

Review History

RSOS-192092.R0 (Original submission)

Review form: Reviewer 1

Is the manuscript scientifically sound in its present form?

No

Are the interpretations and conclusions justified by the results?

No

Is the language acceptable?

Yes

Do you have any ethical concerns with this paper?

No

Have you any concerns about statistical analyses in this paper?

No

Recommendation?

Major revision is needed (please make suggestions in comments)

Comments to the Author(s)

This work concerning the new method for measuring CMC value. This manuscript previously was submitted to RSC Advances, and I was one of the referees. The former version of this manuscript was rejected by me, because of the applicability of analytical method. The authors mentioned something like “which demanded the same fluorospectrometer, the same excitation and emission slit widths of the same fluorospectrometer and even the same measured number for sample measurement.” I believed that this type of new method for measuring CMC value is useless.

For this revised manuscript submitted to Royal Society Open Science, the paragraph like “which demanded the same fluorospectrometer, the same excitation and emission slit widths of the same fluorospectrometer and even the same measured number for sample measurement.” has been deleted by the authors. However, nearly all the data included in the revised manuscript are same as that of in previous paper. Therefore, the authors should demonstrate the reliability of their methods with a point by point response to the reviewer's comments, before this paper can be published.

My reports for the former version of this manuscript was copied below:

In this manuscript, depending on their previously invented sample-preparation method (III), the authors are trying to develop a new method for measuring CMC value by using pyrene as probe. Although quite a lot of data were included, this work still cannot be published in present state. The reasons are as following:

- 1, A useful analytical method should have better applicability. Obviously, the things like “which demanded the same fluorospectrometer, the same excitation and emission slit widths of the same fluorospectrometer and even the same measured number for sample measurement.” are completely useless.
- 2, The authors should explain why the surfactant SDS or CTAB from different suppliers have different CMC values. Is there any impurity?
3. The editing of the paper is very careless. For example, “The main text of the article should appear here with headings as appropriate.” appears at the beginning of main text. And there is “Bibliography” before each reference.

Review form: Reviewer 2

Is the manuscript scientifically sound in its present form?

Yes

Are the interpretations and conclusions justified by the results?

Yes

Is the language acceptable?

Yes

Do you have any ethical concerns with this paper?

Yes

Have you any concerns about statistical analyses in this paper?

Yes

Recommendation?

Accept with minor revision (please list in comments)

Comments to the Author(s)

Critical micelle concentration (CMC) is an important parameter of surfactants with wide applications. The determination of CMC using fluorescence dyes should be a general method. In this work, the authors compared the conventional methods (I and II) and the method (III) developed for CMC probes with aggregation-induced emission (AIE) characteristics. This work is well conducted and organized. This topic is interest and should be of broad audience. I recommend the acceptance after the following issues could be addressed.

1. The determination of CMC of surfactants using fluorescence has been extensively investigated previously. Why the authors choose to use the AIEgens to determine the CMC.
2. The advantages using AIEgens to determine the CMC of surfactants should be clearly described during revisions.
3. The AIE effect has also be utilized for determination of the CMC of AIEgens-containing amphiphilic copolymers. I suggest the authors add some contexts about the AIEgens-containing copolymers and their applications. Some related reviews and reports (e.g., *Chemical reviews* 109 (11), 5799-5867, *Applied Materials Today* 9, 145-160, *Dyes and Pigments* 148, 52-60, *Materials Science and Engineering: C* 81, 416-421, *Materials Science and Engineering: C* 80, 708-714, *Materials Science and Engineering: C* 80, 578-583, *Materials Science and Engineering: C* 80, 411-416, *Materials Science and Engineering: C* 79, 563-569, *Materials Science and Engineering: C* 79, 590-595, *Materials Science and Engineering: C* 78, 862-867, *Chemical Engineering Journal* 308, 527-534, *Polymer Chemistry* 8 (37), 5644-5654, *Materials Science and Engineering: C* 66, 215-220, *Materials Science and Engineering: C* 94, 270-278, *Journal of colloid and interface science* 519, 137-144, *Chemical Engineering Journal* 337, 82-89, *Journal of colloid and interface science* 513, 198-204, *Nanoscale* 7 (27), 11486-11508, *Polymer Chemistry* 5 (2), 356-360, *Polymer Chemistry* 5 (2), 399-404.) should be mentioned and cited during revisions.

Decision letter (RSOS-192092.R0)

13-Jan-2020

Dear Professor Zhu:

Title: Influent factors on pyrene-based CMC determination and a simple method of preparing samples

Manuscript ID: RSOS-192092

The editor assigned to your manuscript has now received comments from reviewers. We would like you to revise your paper in accordance with the referee and Subject Editor suggestions which can be found below (not including confidential reports to the Editor). Please note this decision does not guarantee eventual acceptance.

Please submit your revised paper before 05-Feb-2020. Please note that the revision deadline will expire at 00.00am on this date. If we do not hear from you within this time then it will be assumed that the paper has been withdrawn. In exceptional circumstances, extensions may be possible if agreed with the Editorial Office in advance. We do not allow multiple rounds of revision so we urge you to make every effort to fully address all of the comments at this stage. If

deemed necessary by the Editors, your manuscript will be sent back to one or more of the original reviewers for assessment. If the original reviewers are not available we may invite new reviewers.

RSC Associate Editor:
Comments to the Author:
(There are no comments.)

RSC Subject Editor:
Comments to the Author:
(There are no comments.)

Reviewers' Comments to Author:
Reviewer: 1

Comments to the Author(s)

This work concerning the new method for measuring CMC value. This manuscript previously was submitted to RSC Advances, and I was one of the referees. The former version of this manuscript was rejected by me, because of the applicability of analytical method. The authors mentioned something like "which demanded the same fluorospectrometer, the same excitation and emission slit widths of the same fluorospectrometer and even the same measured number for sample measurement." I believed that this type of new method for measuring CMC value is useless.

For this revised manuscript submitted to Royal Society Open Science, the paragraph like “which demanded the same fluorospectrometer, the same excitation and emission slit widths of the same fluorospectrometer and even the same measured number for sample measurement.” has been deleted by the authors. However, nearly all the data included in the revised manuscript are same as that of in previous paper. Therefore, the authors should demonstrate the reliability of their methods with a point by point response to the reviewer's comments, before this paper can be published.

My reports for the former version of this manuscript was copied below:

In this manuscript, depending on their previously invented sample-preparation method (III), the authors are trying to develop a new method for measuring CMC value by using pyrene as probe. Although quite a lot of data were included, this work still cannot be published in present state. The reasons are as following:

- 1, A useful analytical method should have better applicability. Obviously, the things like “which demanded the same fluorospectrometer, the same excitation and emission slit widths of the same fluorospectrometer and even the same measured number for sample measurement.” are completely useless.
- 2, The authors should explain why the surfactant SDS or CTAB from different suppliers have different CMC values. Is there any impurity?
3. The editing of the paper is very careless. For example, “The main text of the article should appear here with headings as appropriate.” appears at the beginning of main text. And there is “Bibliography” before each reference.

Reviewer: 2

Comments to the Author(s)

Critical micelle concentration (CMC) is an important parameter of surfactants with wide applications. The determination of CMC using fluorescence dyes should be a general method. In this work, the authors compared the conventional methods (I and II) and the method (III) developed for CMC probes with aggregation-induced emission (AIE) characteristics. This work is well conducted and organized. This topic is interest and should be of broad audience. I recommend the acceptance after the following issues could be addressed.

1. The determination of CMC of surfactants using fluorescence has been extensively investigated previously. Why the authors choose to use the AIEgens to determine the CMC.
2. The advantages using AIEgens to determine the CMC of surfactants should be clearly described during revisions.
3. The AIE effect has also be utilized for determination of the CMC of AIEgens-containing amphiphilic copolymers. I suggest the authors add some contexts about the AIEgens-containing copolymers and their applications. Some related reviews and reports (e.g., Chemical reviews 109 (11), 5799-5867, Applied Materials Today 9, 145-160, Dyes and Pigments 148, 52-60, Materials Science and Engineering: C 81, 416-421, Materials Science and Engineering: C 80, 708-714, Materials Science and Engineering: C 80, 578-583, Materials Science and Engineering: C 80, 411-416, Materials Science and Engineering: C 79, 563-569, Materials Science and Engineering: C 79, 590-595, Materials Science and Engineering: C 78, 862-867, Chemical Engineering Journal 308, 527-534, Polymer Chemistry 8 (37), 5644-5654, Materials Science and Engineering: C 66, 215-220, Materials Science and Engineering: C 94, 270-278, Journal of colloid and interface science 519, 137-144, Chemical Engineering Journal 337, 82-89, Journal of colloid and interface science 513, 198-204, Nanoscale 7 (27), 11486-11508, Polymer Chemistry 5 (2), 356-360, Polymer Chemistry 5 (2), 399-404.) should be mentioned and cited during revisions.

Author's Response to Decision Letter for (RSOS-192092.R0)

See Appendix A.

RSOS-192092.R1 (Revision)

Review form: Reviewer 2

Is the manuscript scientifically sound in its present form?

Yes

Are the interpretations and conclusions justified by the results?

Yes

Is the language acceptable?

Yes

Do you have any ethical concerns with this paper?

No

Have you any concerns about statistical analyses in this paper?

Yes

Recommendation?

Accept as is

Comments to the Author(s)

Please accept as is.

Decision letter (RSOS-192092.R1)

30-Jan-2020

Dear Professor Zhu:

Title: Influential factors on pyrene-based CMC determination and a simple method of preparing samples

Manuscript ID: RSOS-192092.R1

It is a pleasure to accept your manuscript in its current form for publication in Royal Society Open Science. The chemistry content of Royal Society Open Science is published in collaboration with the Royal Society of Chemistry.

RSC Associate Editor:
Comments to the Author:
(There are no comments.)

RSC Subject Editor:
Comments to the Author:
(There are no comments.)

Reviewer(s)' Comments to Author:
Reviewer: 2

Comments to the Author(s)
Please accept as is.

Appendix A

January 20, 2020

Re: Manuscript ID RSOS-192092

Dear Editor Smith:

Thank you very much for further considering our revised manuscript version entitled “Influent factors on pyrene-based CMC determination and a simple method of preparing samples” for publication in *Royal Society Open Science*.

We also thank the reviewers for their constructive comments and suggestions. We have revised the manuscript accordingly. The attached are our item-by-item responses to the reviewers’ comments and suggestions.

We believe that we have satisfactorily addressed the concerns of the reviewers and hope that the revised manuscript will now be suitable for publication in the *Royal Society Open Science*.

Sincerely yours,

Qihua Zhu

Qihua Zhu, Prof. Ph.D.
School of Pharmaceutical Sciences
Southern Medical University
1838 Guangzhou Avenue North
Guangzhou 510515, China
Tel: +86-20-61647627
E-mail: zhuqh@smu.edu.cn

Responses to reviewers:

Reviewer: 1

Comments to the Author(s)

This work concerning the new method for measuring CMC value. This manuscript previously was submitted to RSC Advances, and I was one of the referees. The former version of this manuscript was rejected by me, because of the applicability of analytical method. The authors mentioned something like “which demanded the same fluorospectrometer, the same excitation and emission slit widths of the same fluorospectrometer and even the same measured number for sample measurement.” I believed that this type of new method for measuring CMC value is useless.

For this revised manuscript submitted to Royal Society Open Science, the paragraph like “which demanded the same fluorospectrometer, the same excitation and emission slit widths of the same fluorospectrometer and even the same measured number for sample measurement.” has been deleted by the authors. However, nearly all the data included in the revised manuscript are same as that of in previous paper. Therefore, the authors should demonstrate the reliability of their methods with a point by point response to the reviewer's comments, before this paper can be published.

Response: Thank you very much for your professional comments and suggestions.

My reports for the former version of this manuscript was copied below:

In this manuscript, depending on their previously invented sample-preparation method (III), the authors are trying to develop a new method for measuring CMC value by using pyrene as probe. Although quite a lot of data were included, this work still cannot be published in present state. The reasons are as following:

1, A useful analytical method should have better applicability. Obviously, the things like “which demanded the same fluorospectrometer, the same excitation and emission slit widths of the same fluorospectrometer and even the same measured number for sample measurement.” are completely useless.

Response: Thank you very much for your question.

Yes, a useful analytical method should have better applicability. The mentioned new method III for preparing samples for CMC determination in the manuscript does not require the same fluorospectrometer, the same excitation and emission slit widths of the same fluorospectrometer and even the same measured number for sample measurement. It is the I_{FIII}/I_{FI} value of pyrene that is unexpectedly sensitive to the measurement conditions relating to the light energy exciting pyrene such as slit widths and measurement number.

Generally, for organic fluorescent compounds, their fluorescence intensities change with measured conditions, but the ratios of their fluorescence intensities at two wavelengths usually depend on its nature and environment rather than measured conditions. When we prepared surfactant samples containing pyrene by conventional method I and determined the CMC value of SDS by measuring

the $I_{\text{FIII}}/I_{\text{FI}}$ value of pyrene, we found that the $I_{\text{FIII}}/I_{\text{FI}}$ value sometimes varied irregularly. This puzzled us and then we explored the factors influencing $I_{\text{FIII}}/I_{\text{FI}}$ value in detail. We found that the $I_{\text{FIII}}/I_{\text{FI}}$ value of pyrene is unusually sensitive to the measurement conditions changing exciting light energy. This is very important and useful for decreasing and understanding the measured errors of $I_{\text{FIII}}/I_{\text{FI}}$ value and obtaining more precise CMC values (the reported CMC errors of SDS determined using pyrene as probe is high to ± 0.4 mM in *J. Fluoresc.*, 2018, **28**, 465-476).

We have revised "the $I_{\text{FIII}}/I_{\text{FI}}$ value of pyrene, which demanded the same fluorospectrometer, the same excitation and emission slit widths of the same fluorospectrometer and even the same measured number for sample measurement" as "the $I_{\text{FIII}}/I_{\text{FI}}$ value of pyrene is unusually sensitive to the measurement conditions relating to the light energy exciting pyrene, such as slit widths and measurement number."

2, The authors should explain why the surfactant SDS or CTAB from different suppliers have different CMC values. Is there any impurity?

Response: Thanks very much for your professional suggestion.

As you indicated, the impurity shows great influence on the CMC value (*J Fluoresc* 2018, **28**(1): 465-476¹). We have mentioned the influence of impurity in the revised manuscript.

3. The editing of the paper is very careless. For example, "The main text of the article should appear here with headings as appropriate." appears at the beginning of main text. And there is "Bibliography" before each reference.

Response: We are very sorry for this. We usually prepared and checked the manuscript in a Microsoft Word format, and then copied it to a journal template. We should check our manuscript carefully after copying the prepared manuscript to a Magazine template. The word "Bibliography" before each reference is automatically generated by the endnote software. It was the first time to encounter this problem. We have carefully checked our manuscript and deleted this in the revised manuscript.

Reviewer: 2

Comments to the Author(s)

Critical micelle concentration (CMC) is an important parameter of surfactants with wide applications. The determination of CMC using fluorescence dyes should be a general method. In this work, the authors compared the conventional methods (I and II) and the method (III) developed for CMC probes with aggregation-induced emission (AIE) characteristics. This work is well conducted and organized. This topic is interesting and should be of broad audience. I recommend the acceptance after the following issues could be addressed.

Response: Thank you very much for your positive comments.

1. The determination of CMC of surfactants using fluorescence has been extensively investigated previously. Why the authors choose to use the AIEgens to determine the CMC.

Response: Thank you very much for your good question.

Although the determination of CMC of surfactants using fluorescent probes has been extensively investigated previously, the procedures of determining CMC values are still tedious, time- and sample-consuming. This is because the fluorescence method is based on the fluorescence change of probes with surfactant concentrations and the fluorescent changes are usually invisible or not sharp about CMC. Hence, a series of samples containing different concentrations of surfactant and a certain amount of probe are needed to be prepared and measured by a fluorospectrophotometer. If the fluorescence changes are very sharp and visible about/at CMC, the CMC values will be directly observed. Therefore, highly sensitive and clearly visible fluorescence probes are desired in practical applications.

Considering AIEgens show strong fluorescence in aggregates but no emission in monomers, we thought that AIEgens might be developed as such highly sensitive and visible fluorescence-turn-on probes for CMC observation. As we expected, AIE THPs, synthesized by a convenient multi-component reaction that we developed, can be used highly sensitive and visible fluorescence-turn-on (showing the strongest fluorescence at CMC) probes for CMC determination of ionic surfactants (*Chem. Commun.*, 2014, **50**, 1107-1109). Recently, we found that one of THPs could be used as an excellent indicator for CMC titration and realized simple, sample- and time-saving CMC titration for different kinds of surfactant for the first time (*Anal. Chem.*, 2019, 10.1021/acs.analchem.9b04638).

It is worth mentioning that the reported CMC fluorescent probes, including the reported AIE probe (*Analyst* **2011**, *136*, 3343²), based on fluorescence intensity change are all fluorescence-turn-off probes, that is, showing the weakest fluorescence at CMC.

2. The advantages using AIEgens to determine the CMC of surfactants should be clearly described during revisions.

Response: Thank you very much for your suggestion.

I have mentioned the advantages using AIEgens to determine the CMC of surfactants in the

introduction section (please see the revised manuscript).

3. The AIE effect has also be utilized for determination of the CMC of AIEgens-containing amphiphilic copolymers. I suggest the authors add some contexts about the AIEgens-containing copolymers and their applications. Some related reviews and reports (e.g., Chemical reviews 109 (11), 5799-5867, Applied Materials Today 9, 145-160, Dyes and Pigments 148, 52-60, Materials Science and Engineering: C 81, 416-421, Materials Science and Engineering: C 80, 708-714, Materials Science and Engineering: C 80, 578-583, Materials Science and Engineering: C 80, 411-416, Materials Science and Engineering: C 79, 563-569, Materials Science and Engineering: C 79, 590-595, Materials Science and Engineering: C 78, 862-867, Chemical Engineering Journal 308, 527-534, Polymer Chemistry 8 (37), 5644-5654, Materials Science and Engineering: C 66, 215-220, Materials Science and Engineering: C 94, 270-278, Journal of colloid and interface science 519, 137-144, Chemical Engineering Journal 337, 82-89, Journal of colloid and interface science 513, 198-204 , Nanoscale 7 (27), 11486-11508, Polymer Chemistry 5 (2), 356-360, Polymer Chemistry 5 (2), 399-404.) should be mentioned and cited during revisions.

Response: I have read the references on the preparation methods and applications of AIEgens-containing amphiphilic copolymers. These works are excellent and I will explore the preparation of THP-containing amphiphilic copolymers and their applications. Thank you very much. I have added several of these references in the introduction section of the revised manuscript.